# Quantifying the Influence of Pollen Aging on the Adhesive Properties of *Hypochaeris radicata* Pollen

**DOI:** 10.3390/insects13090811

**Published:** 2022-09-06

**Authors:** Steven Huth, Lisa-Maricia Schwarz, Stanislav N. Gorb

**Affiliations:** 1Zoological Institute, Kiel University, Am Botanischen Garten 1-9, 24118 Kiel, Germany; 2Grassland Ecology & Grassland Management, Department of Plant Nutrition, Institute of Crop Science and Resource Conservation (INRES), University of Bonn, Karlrobert-Kreiten-Str. 13, 53115 Bonn, Germany; 3Biodiversity Research and Systematic Botany, Institute of Biochemistry and Biology, University of Potsdam, Maulbeerallee 1, 14469 Potsdam, Germany

**Keywords:** pollination, pollen adhesion, pollenkitt, atomic force microscopy, cryogenic scanning electron microscopy, centrifugation

## Abstract

**Simple Summary:**

Pollination is the transfer of pollen from a plant’s male part (anther) to the corresponding female part (stigma). It is a fundamental biological process that ensures plant reproduction. Most studies investigate pollination from a biological perspective, but the underlying physical processes are poorly understood. Many plants rely on insects to transport pollen and the forces with which pollen adhere to insects and floral surfaces are fundamental for successful pollination. We quantified pollen adhesion by measuring the forces necessary to detach *Hypochaeris radicata* (catsear, a common insect-pollinated plant) pollen from glass and studied for the first time how the adhesion forces change with pollen aging. Our results show that newly formed adhesion bonds between *H. radicata* pollen and glass are stronger for fresh pollen than for old ones. On the other hand, when *H. radicata* pollen age in contact with glass, the adhesion between pollen and glass strengthens over time. These effects are probably caused by the viscous liquid covering most pollen (pollenkitt) changing its viscoelastic properties as it dries.

**Abstract:**

Although pollination is one of the most crucial biological processes that ensures plant reproduction, its mechanisms are poorly understood. Especially in insect-mediated pollination, a pollen undergoes several attachment and detachment cycles when being transferred from anther to insect and from insect to stigma. The influence of the properties of pollen, insect and floral surfaces on the adhesion forces that mediate pollen transfer have been poorly studied. Here, we investigate the adhesive properties of *Hypochaeris radicata* pollen and their dependence on pollen aging by quantifying the pull-off forces from glass slides using centrifugation and atomic force microscopy. We found that the properties of the pollenkitt—the viscous, lipid liquid on the surface of most pollen grains—influences the forces necessary to detach a pollen from hydrophilic surfaces. Our results show that aged *H. radicata* pollen form weaker adhesions to hydrophilic glass than fresh ones. On the other hand, when a pollen grain ages in contact with glass, the adhesion between the two surfaces increases over time. This study shows for the first time the pollen aging effect on the pollination mechanism.

## 1. Introduction

Pollination is a highly efficient and species-specific biological particle transport system that forms the basis for plant reproduction and distribution [1,2]. It is fundamental for our planet’s ecosystem and for human food production, and it plays an important role in the mediation of atmospheric phenomena [3,4,5]. Despite the fact that the principle of pollination is common knowledge, remarkably little is known about the details of pollen transfer mechanisms [6]. Most studies have focused on pollination ecology, chemical recognition mechanisms, germination or other biochemical aspects of pollination [1,3,7,8,9,10,11]. However, adhesion processes play a crucial role during pollen transfer from the anther to the stigma. Especially with insect-dependent pollination, pollen undergo multiple attachment and detachment cycles to surfaces with various chemical and structural features [5]. In order to facilitate efficient pollen transfer, floral and pollen surfaces must present a specific set of properties that passively influence adhesion forces. These adhesion forces cause the pollen to be transferred from the flower anther to the insect body and eventually from the insect to the flower stigma. Therefore, adhesive mechanisms, as well as the influence of floral and insect surface properties on adhesion forces, are an essential part of the pollination process.

This has only been recently addressed by very few studies [2,5,6,12]. These studies have shown that pollen adhesion dynamics are affected by an interplay of surface structures and liquids that are often present on floral surfaces. For instance, pollenkitt—the surface liquid covering pollen grains—is widely accepted to increase pollen adhesion [10,13,14]. On the other hand, it has recently been shown that pollenkitt may reduce adhesion to wet surfaces [6], suggesting that its role during pollination is more complex than widely assumed. It has also been stated that the pollenkitt of most pollen partially dries out during the transfer process from the anther to the stigma [10,15].

To the best of our knowledge, the adhesion properties of pollen as well as the influence of pollen aging and dehydration have not yet been quantified experimentally. It is plausible to hypothesize that pollenkitt, which is the most outer layer of pollen grains, is influenced by dehydration and therefore by the time that has passed since the pollen presentation on the anther. Here, we study how the adhesion properties of *Hypochaeris radicata* pollen change over time by quantifying the forces necessary to detach pollen from glass surfaces using centrifugation and atomic force microscopy (AFM). Centrifugation allows the quantification of pollen adhesion for many pollen grains in one experiment and therefore offers the opportunity to draw statistically relevant conclusions. AFM experiments, on the other hand, quantify the adhesion of single pollen grains. These experiments are repeatable and therefore allow us to examine one and the same pollen individual at different experimental and environmental conditions with highly comparable results. This opens the possibility to realize very detailed studies on the influence of a variety of factors on pollen adhesion. Therefore, such a detailed AFM analysis is complementary to the statistically relevant centrifugation approach. We also employed cryo-scanning electron microscopy (cryo-SEM) to observe pollen grains in their native conditions with their surface liquids present.

In this study, we examine the change in morphology of *H. radicata* pollen grains due to aging and the impact this has on their adhesive bonds’ properties as well as their ability to form new adhesions to hydrophilic surfaces. We also studied the dependence of pull-off forces necessary to detach an individual *H. radicata* pollen grain from a hydrophilic glass surface on both the contact force and the detachment speed.

A deeper understanding of the pollination process will provide insights into bioinspired passive microparticle manipulation through the altering of surface properties, which is relevant for artificial pollination, drug delivery [16], respiratory medicine [17], and the engineering of biomimetic microgripping systems [2].

## 2. Materials and Methods

### 2.1. Pollen Sample Preparation

The flowering stems of *Hypochaeris radicata* (Asteraceae) were collected from open spaces around Kiel University (Kiel, Germany) between July and November. To preserve the functionality of the collected flowering stems in the laboratory, they were kept in water. Ray flowers that had pushed their styles through the anthers were chosen and by carefully brushing an eyelash over the styles, the pollen grains were removed and poured onto cleaned glass coverslips. Using an ultrasonic bath, glass slides were cleaned for 5 min in 70% ethanol and subsequently for 5 min in double-distilled water. They were blown dry with compressed air. To analyze pollen without pollenkitt, the latter was removed by washing the pollen grains ten times in a mixture of chloroform and methanol (3:1). The solvent containing the pollenkitt was filtered out between washing procedures.

### 2.2. Centrifugation

For centrifugation assays, glass slides were cut into pieces fitting a 2.5 mL Eppendorf tube, cleaned and populated with pollen grains. The bottom of each glass piece was marked for target regions using 8 mm × 6 mm grids. Only target regions that contained at least 20 pollen grains were chosen for analysis. For each target area, an image of the pollen particles was taken using a Leica M205 microscope (Leica Microsystems GmbH, Wetzlar, Germany) before and after the sample was centrifuged for 3 min at 1000 rpm in a Heraeus Sepatech Biofuge A table centrifuge (Heraeus Holding GmbH, Hanau, Germany). The experiment was repeated 13 times with increasing rotation speeds (1000 rpm, 2000 rpm, 3000 rpm, …, 13,000 rpm). The number of single pollen grains on each of the resulting images was counted using a home-written Matlab (MathWorks, Natick, US) algorithm. Clumps of pollen grains were excluded.

The centrifugal force is defined as $F=m\omega^{2}r$ with *m* being the mass of a single pollen, ω the angular velocity and *r* the radius of rotation. Therefore, the mass of pollen particles was necessary to obtain the centrifugal force. To determine the pollen mass, we first weighed approximately 1000 pollen grains with a UMX2 Ultra-microbalance (Mettler Toledo, Columbus, USA). These grains were subsequently spread on a glass slide to ensure that they were positioned in a single layer. We then took a light microscopy image of the monolayer. The customized Matlab particle counting algorithm was used to determine the exact number of pollen grains per sample, which was, in turn, used to calculate the average mass of a single pollen grain. We repeated the experiment ten times for fresh and 7-day old *H. radicata* pollen, respectively.

### 2.3. Cantilever Preparation

We calibrated AFM cantilevers (HQCSC37/NO_AL, MikroMasch OÜ, Tallinn, Estonia) with a Nanowizard I AFM (JPK, Berlin, Germany). The sensitivity was determined by collecting 10 force–distance curves with a setpoint of 1 V at a speed of 10 µm/s on each of 16 different positions on a cleaned glass slide. Subsequently, the spring constant was calibrated ten times using the well-established thermal noise method and the mean value was used for the experiments. The cantilevers used in this study had spring constants ranging from 0.3 N/m to 0.5 N/m.

A single *Hypochaeris radicata* pollen particle on a glass slide was chosen and a small amount of 2-component epoxy adhesive (UHU Schnellfest, UHU GmbH & Co. KG, Bühl, Germany) was deposited next to it. A calibrated cantilever was approached so that the tip was dipped into the glue. Afterwards, residual glue was removed by dipping the cantilever onto the glass once or twice before the cantilever was brought into contact with the pollen particle. The pollen and cantilever were retracted from the glass slide and the glue was allowed to cure for 30 min before the experiments were conducted.

### 2.4. Atomic Force Microscopy

In order to quantify the influence of the setpoint force and cantilever speed on the pull-off forces of a *Hypochaeris radicata* pollen from glass (Figures 4 and 6), a pollen-cantilever was used to collect 10 force–distance curves on each of 9 positions on a cleaned glass slide with the respective experimental parameters. For 0.1 µm/s, only 5 curves were recorded per position. The sampling rates were synchronized so that each curve consisted of 1024 data points. The results were plotted as boxplots using a home-written Python algorithm. Statistical tests were carried out in Python (Python Software Foundation, Beaverton, OR, USA). Since the residuals of our data were not normally distributed, we conducted Kruskal–Wallis tests and Dunn’s post hoc test with Bonferroni correction. For this purpose, we tested the data in groups. For the influence of setpoint force, we tested groups of data collected at the same cantilever speed. Similarly, the influence of cantilever speed was tested by analyzing data with same setpoint force.

The long-time measurements (Figure 5) were realized on a clean glass slide with 1 nN setpoint force. Curves were collected at several positions to exclude artifacts from non-specific pollen–glass interactions. A list of positions was repeatedly measured, recording one curve per position. Per cycle, the mean of all positions was calculated and plotted. For the measurement carried out at 10 µm/s, the curves were collected at 9 different positions with a pause of 300 s after each curve. For 0.1 µm/s, curves were collected at 9 different positions without any pause and for 1 µm/s, curves were collected at 6 different positions with 600 s pause after each curve.

### 2.5. Cryo-Scanning Electron Microscopy (Cryo-SEM)

Opened florets or AFM cantilevers carrying a pollen particle were fixated on SEM stubs. These stubs were subsequently frozen at −140 °C in the Gatan Alto 2500 cryo-preparation system (Gatan, Pleasanton, CA, USA) and sputter coated with 10 nm Au-Pd under frozen conditions so that surface liquids did not evaporate and thus could be visualized. Images were taken at −120 °C and 3 kV accelerating voltage with a Hitachi S 4800 scanning electron microscope (Hitachi High Technology, Tokyo, Japan).

## 3. Results and Discussion

We imaged fresh *Hypochaeris radicata* pollen with and without pollenkitt using cryo-SEM (Figure 1a,b and Figure 1c,d, respectively). Moreover, cryo-SEM images of a *H. radicata* pollen, glued to an AFM cantilever and stored at room temperature for more than 7 days were taken (Figure 1e,f). A pollenkitt layer covers the entire surface of fresh *H. radicata* pollen grains and only the tips of the pollen spines are visible (Figure 1a,b). As the pollenkitt is removed, the pollen surface morphology changes considerably: the pollen show deeper cavities, the rims of which are formed by elevated areas that carry the fully exposed spines (Figure 1c,d). Apparently, most of the pollenkitt is deposited in these cavities, covering the porous surface of the pollen’s outer wall (exine). Old and dehydrated *H. radicata* pollen also show these cavities (Figure 1e,f). However, the pores, despite being clearly visible, are still covered by a thin layer of pollenkitt. Moreover, the aged pollenkitt is more apparent at the cavities’ edges (marked with white arrows) and between the spines, which are not as exposed as they are for pollen grains without pollenkitt. Thus, the pollenkitt on aged pollen still covers the pollen surface, but has been dehydrated and reduced to a very thin layer that renders the morphology to be dominated by the exine.

To study the influence of pollen aging on pollen adhesion, we quantified the adhesive properties of fresh and aged *Hypochaeris radicata* pollen with a centrifugation assay. Pollen adhesion was quantified by measuring the force necessary to detach pollen particles from glass via the centrifugation of a glass slide with adhering *H. radicata* pollen particles at different centrifugation speeds and therefore centrifugal forces. After each centrifugation, the amount of pollen that detached due to the respective centrifugal force was determined. It is important to note that higher centrifugation speeds do not only correspond to higher forces, but also to an extended duration of mechanical load as the centrifuge increases its speed gradually from 0 rpm. Especially for viscous samples, this might result in lower detachment forces compared to instantaneous force application. However, since we employ this method to compare the adhesion of different pollen particles and as these considerations apply to all our samples equally, centrifugation is applicable and our comparative results are valid. The advantage of this type of experiment is that a single experiment results in the adhesion analysis of many pollen particles at the same conditions.

While fresh pollen were directly deposited on the glass slide after being collected from the plant stylus, aged pollen were collected fresh and kept at room temperature for seven days before they were deposited on the glass slide. In both cases, centrifugation experiments were carried out directly after the pollen deposition, which means that short-term adhesive interactions were quantified. Furthermore, the long-term adhesive properties of *H. radicata* pollen were quantified by depositing fresh pollen on a glass slide and allowing them to adhere to it for seven days before the centrifugation experiments were carried out.

The results of the centrifugation experiment are presented in Figure 2. The adhesion of fresh *H. radicata* pollen that aged in contact with glass was much stronger than that of fresh and old pollen that were tested immediately after being deposited on glass. Most fresh and old pollen grains detached from the glass surface at a force range between 10 nN and 100 nN. In both latter cases, less than 30% of pollen remained attached after experiencing a centrifugal force of 100 nN. For the pollen grains aged in contact, on the other hand, there was no critical centrifugal force that removed the majority of pollen grains, but they detached more gradually and at higher forces. After applying a centrifugal force of 831 nN, 40% of these pollen still remained adhering to the glass, while less than 20% of fresh and old pollen withstood this force. Moreover, the results suggest that old *H. radicata* pollen adhere less strongly to glass than fresh ones. The difference is not as pronounced as it is for fresh pollen aged in contact, but for each centrifugal speed, the mean detachment force of old pollen was lower than that of fresh pollen.

One possible explanation of these observations is a change of pollenkitt properties, when the pollen dehydrate. Pollenkitt is largely regarded as a sticky emulsion of water and oil [14] and should thus become denser, more viscous and more adhesive when dehydrated [18]. On the one hand, a stickier pollenkitt may lead to an increase in pollen adhesion, however, the increased viscosity of the pollenkitt, on the other hand, slows down or even inhibits the formation of fluid bridges, which might be crucial for the fast and efficient formation of strong pollen adhesion [6]. This latter effect reduces the contact area—and therefore adhesive interactions—between the pollen particle and the substrate. The contact area is further reduced due to the fact that the pollenkitt layer shrinks with time because of the partial pollenkitt evaporation (Figure 1). Thus, short-term pollen adhesion, which has been analyzed for fresh and old pollen grains immediately after the contact formation with the substrate, is weakened when the pollenkitt evaporates partially. The reason for this is that the higher viscosity of old pollenkitt increases the time that is needed for the pollenkitt to flow onto the substrate to form fluid bridges. Long-term adhesion, which has been quantified for fresh pollen grains after they spent seven days in contact with the substrate, is increased, presumably due to the fact that fluid bridges that had already formed were solidified and strengthened when the pollenkitt became more condensed and stickier after partial evaporation.

To study the effects of pollen aging on the adhesive properties of pollen grains in more detail, we attached single *H. radicata* pollen grains to AFM cantilevers. Figure 1e) shows an exemplary image of an old *H. radicata* pollen glued to an AFM cantilever. It can be seen that the epoxy glue is only present between the pollen and the cantilever with the rest of the pollen surface left pristine during the gluing process. Pollen adhesiveness was quantified by bringing a pollen attached to a calibrated cantilever into contact with glass and by recording cantilever deformation while removing the pollen from the glass surface after the contact was formed. An example of a resulting force–distance curve is presented in Figure 3a. The minimum of the retract segment of the curves corresponds to the force necessary to detach the pollen from the glass slide (“pull-off force”). As this is the force that overcomes the adhesive forces between pollen and glass, we used the pull-off force to quantify pollen adhesion.

In some rare cases, electrostatic forces, visible as long-range interactions in the curves, were recorded. In these cases, we removed the surface charges from pollen and glass slide by employing an ionizing blow-off gun. Figure 3b presents the approach segments of an AFM curve recorded before and after the deionization procedure. The results clearly show that the long-range interactions can be present, but they were effectively removed by deionization.

The advantage of AFM-based adhesion quantification is that the adhesion of one and the same pollen particle can be repeatedly measured and thus, the influence of different experimental and environmental conditions can be analyzed using a single individual. Therefore, many errors originating from biological individuality do not affect such experiments and relevant qualitative conclusions can be drawn even with a relatively low number of individuals. Here, we chose to combine AFM studies with the centrifugation approach in order to have very detailed comparative results on the one hand and a high amount of individuals for quantitative results on the other. In order to ensure the repeatability of the AFM experiments, we chose the setpoint forces, which correspond to the force that the pollen experiences in contact with the glass slide, sufficiently low so that the pollen is not deformed during the experiments. As long as the pollen is undeformed, the part of the force–distance curve that is recorded during pollen–glass contact (marked with the dashed oval in Figure 3a) remains linear. For each experiment, we collected force–distance curves at 6–9 different positions on the glass slide to avoid artifacts from unspecific interactions between pollen grains and glass substrate. Furthermore, we repeatedly measured the pull-off forces of a single pollen particle 320 times and could not find a significant change in pollen adhesion (Figure 3c).

First, we tested the influence of the setpoint force and cantilever speed on the measured pull-off forces of single *H. radicata* pollen from glass in order to optimize these experimental parameters for our measurements. Figure 4a,b present the results obtained for two individual pollen grains. Moreover, we plotted the pull-off forces versus the measurement number for each pollen grain (Figure 4c,d) to ensure that our results were not influenced by any artifacts due to repeated measurements. For both individual pollen grains, the pull-off forces increased with the cantilever speed (pull-off rate), which is probably caused by the viscosity of the pollenkitt. Interestingly, at low cantilever speeds of 0.1 µm/s, both pollen grains had similar pull-off forces of approximately 50 nN. At higher cantilever speeds, the pull-off forces increased more strongly for the first pollen than for the second one. Thus, lower pull-off rates resulted in more comparable measurements between individual pollen grains.

On the other hand, the pull-off forces of the first pollen continuously decreased over time when measured at low cantilever speeds of 0.1 µm/s or 1 µm/s (Figure 4c). At higher speeds, the measurements are repeatable. This did not only result in broader distributions of pull-off force values, but it is also a possible explanation for the fact that the influence of the setpoint forces on the pull-off forces is much more pronounced for low pull-off rates (Figure 4b). At higher cantilever speeds of 10 µm/s and 25 µm/s, the pull-off forces are not strongly influenced by setpoint forces between 0.5 nN and 25 nN.

This might be due to the fact that pollen aging influences our results, which should be more pronounced for slow cantilever speeds, as more time passes between consecutive measurements. Another possible explanation could be that slow cantilever speeds give the pollenkitt more time to form contact with the glass, which would increase the probability of some pollenkitt residues to remain on the glass when the pollen is removed. However, pollenkitt mostly consists of lipids, and is therefore a hydrophobic substance [14]. Glass, on the other hand, is hydrophilic and thus, an increased amount of pollenkitt on the glass should rather strengthen adhesion instead of weakening it. The second pollen did not show such a behavior, so this effect either might be due to the individual differences between pollen grains, their possible different ages when collected from the flower, or because the limits of repeatability were reached for the first pollen. In any case, it is important to double-check the results with such consecutive plots and to choose the pull-off rate carefully when conducting experiments. Lower pull-off rates ensure comparability between individual pollen grains but higher pull-off rates, on the other hand, decrease the risk of artifacts when a single pollen grain is measured repeatedly.

In order to study the influence of pollen age on the adhesive properties of a pollen grain, we repeatedly measured the pull-off force of three *H. radicata* pollen from glass for several hours. We carried out the experiments with different cantilever speeds to further study the mechanisms that make pollen adhesion dependent on pull-off rate. The results show that all three pollen decrease their adhesion over time (Figure 5). After the first pollen showed a continuous decrease in pull-off forces up to 16.2% from 108.5 nN to 90.8 nN over 12 h (Figure 5a), we increased the measurement time to 53 h for the next pollen (Figure 5b). For this pollen, the pull-off forces increased from 48.3 nN to 51.5 nN during the first hour and then decreased for 25% to 38.5 nN after 35 h, after which it did not change strongly anymore. The data for the third pollen show the decrease in pull-off forces by 10.4% from 65.7 nN to 58.9 nN in 26 h, after which the force decreased further after the measurement site on the glass slide was changed after 48 h (Figure 5c). Figure 5d shows the pull-off forces of all three pollen grains over time.

In addition to the fact that all pollen decreased their adhesiveness, other observations are worth noticing: For instance, we used a higher measurement frequency for the second pollen (Figure 5b) in order to test if the measured decrease in pull-off forces originates from the aging of pollen or from the pollen surface wear from measuring repeatedly. Apart from an initial increase, the pull-off forces of this pollen decreased over time in a comparable manner as for the other individual pollen grains. As this is the case despite the higher amount of collected force–distance curves, we conclude that our results indeed mainly originate from the effects of pollen aging instead of the pollen surface wear.

The initial increase in pull-off forces by 3.2 nN recorded for the second pollen (Figure 5b) might be caused by natural fluctuations of the results, which are due to the complexity of the pollen–glass interaction. The results presented in Figure 3c also show force fluctuations of this magnitude. Another possible explanation might be that very young pollen increase their adhesiveness for a short time, before the decrease begins. Gluing a pollen to a cantilever includes a curing time of 30 min. Moreover, individual pollen grains might have already different ages when they are glued to the cantilever, because the time at which a blossom already blooms before pollen collection varies as well. A combination of these two factors might lead to the situation that most individual pollen grains have already passed the time of increasing their adhesiveness before we started our experiment. The proof of this hypothesis would be a very interesting topic for future studies.

It is striking that the decrease in pull-off forces ceases for the second and third pollen after 35 h and 26 h, respectively. This might either be caused by the pollen itself, which might stop changing its adhesive properties, or it might be due to an assimilation of hydrophobic pollenkitt residuals on the hydrophilic glass slide, which would increase pollen adhesion. Indeed, the pull-off forces of the second and third pollen increased slightly after having reached a minimum. If the pollen continues to decrease its adhesiveness, these two effects could cancel each other out, which would explain the plateaus in our force data. The fact that the adhesion of the third pollen decreased further after we changed to a different position on the glass slide, is a confirmation of that hypothesis. On the other hand, the second pollen should show a similar effect much earlier, as we quantified its adhesion over time with a higher measurement frequency.

Figure 5d shows the different pull-off force ranges for the three individual pollen grains presented in Figure 5a–c. This is in accordance with the speed dependency which we presented in Figure 4. Interestingly, the relatively high adhesion of pollen a, measured at 10 µm/s, decreased the fastest. A possible explanation for this result is that the pollenkitt becomes more viscous over time. Even though a higher pollenkitt viscosity should theoretically increase the pull-off forces for high cantilever speeds, it also increases the time that the pollenkitt needs to form fluid bridges with the glass slide. As higher cantilever speeds result in less contact time, an increased viscosity could actually weaken short-term adhesion.

It is also very interesting that the third pollen measured at 1 µm/s only very slightly decreases its adhesion. Possible reasons are individual pollen characteristics or a seasonal effect, as the pollen grains were harvested at different times of the year (August and November). Whether the season does influence pollen adhesiveness remains unclear and would be a very interesting topic for future studies.

We further tested the hypothesis that pull-off forces are mostly affected by the change of pollenkitt viscosity when the pollen particle ages by measuring the influence of setpoint forces and cantilever speeds for two *H. radicata* pollen older than seven days (Figure 6).

Strikingly, the influence of these two experimental parameters is very different than for fresh pollen. With few exceptions, higher cantilever speeds now result in lower pull-off forces. This fact is in accordance with our hypothesis that the higher viscosity of the pollenkitt increases with time, since this extends the time that the pollenkitt needs to form fluid bridges with the glass. Higher cantilever speeds result in a lower contact time and thus, the pollen adhesion decreases. Interestingly, for speeds above 0.1 µm/s, the pull-off forces of both aged pollen increase with setpoint force. Higher contact forces might increase the contact area between pollenkitt and glass, thus supporting the formation of fluid bridges. However, it is surprising that no similar effect was observed for fresh pollen (Figure 4). Another possible explanation is that during this experiment, pull-off forces increased when measured repeatedly, as shown by Figure 6c,d. Thus, higher setpoint forces, which were measured later, present higher values. This is an interesting observation in itself, as this was not observed for fresh pollen. Another possible reason for the different behavior of the pollen presented in Figure 4 and Figure 6 might be that the former were harvested in July and August, while the latter were collected in November. It remains unclear whether flowers produce dissimilar pollen at different seasons and this would be a very interesting topic for future studies.

It would be interesting to repeat the experiments on hydrophobic surfaces, as pollen grains have to be transferred from hydrophilic plant surfaces to hydrophobic insect surfaces and vice versa during pollination. The pollen age (or the state of pollen evaporation) might be a factor regulating the adhesion forces during such transfer events.

During this study, we removed the surface charges from the pollen. However, it has been hypothesized that electrostatic interactions play an important role during pollination, especially for wind pollinated species [19,20,21,22]. Thus, in future studies, it would be very interesting to analyze the contribution of electrostatic forces to the adhesion of pollen in plant species with different pollination biology.

Another point is that the precise pollen age is not easily determined, as the exact moment the stylus pushes through the anther to present the pollen is hard to define. However, our results suggest that pollen properties might already change within an hour (Figure 5). Moreover, as it takes 30 min for the epoxy to cure when a pollen grain is glued to a cantilever, our method might not be able to quantify the initial adhesive properties of pollen grains. Novel AFM techniques, such as fluidic force microscopy, can potentially be employed to instantly attach a pollen grain to a cantilever with pressurized airflow to overcome this issue.

Furthermore, in order to understand the mechanics of pollination, the adhesion of pollen to more biologically relevant surfaces such as anther, stigma and insects, has to be studied in the future. As adhesion is mediated by a complex interplay of physical forces, which in turn are influenced by a variety of properties of several surfaces, these processes need to be disentangled by systematic research. The study of pollen adhesiveness to glass as a model surface is a first important step and the knowledge gained from this and similar future studies is crucial for the interpretation of more complex experimental designs.

## 4. Conclusions

After we visualized the different appearances of fresh and old *Hypochaeris radicata* pollen using cryo-SEM, we quantified the adhesion of *H. radicata* pollen and the influence of pollen age on pollen adhesiveness with a combination of centrifugation assays and AFM measurements. Our data indicate that pollen decrease their short-term adhesion and increase their long-term adhesion to hydrophilic surfaces over time. We found evidence that the viscosity of the pollenkitt influences the adhesive properties of the pollen. Our data suggest that the change of pollenkitt viscosity over time—which is most likely due to the evaporation of some volatile components—dominates the alteration of pollen adhesiveness. Long-term adhesion is strengthened by this effect, while the formation of new fluid bridges, which are a key factor for initial adhesion formation, is inhibited. This study is the first to investigate the influence of pollen aging on the adhesive properties of pollen grains and is a further step to understand the mechanics of pollination.

## Figures and Tables

**Figure 1 insects-13-00811-f001:**
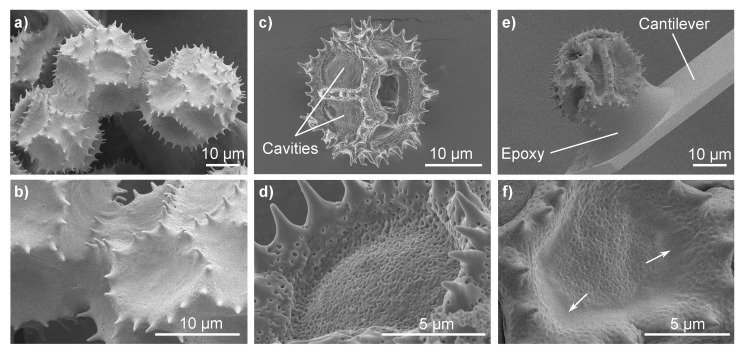
Cryo-scanning electron microscopy (cryo-SEM) images of fresh *Hypochaeris radicata* pollen with pollenkitt present (**a**,**b**) and with pollenkitt removed (**c**,**d**) as well as 7-day-old *H. radicata* pollen (**e**,**f**). The white arrows in (**f**) indicate remnants of pollenkitt on the surface of aged pollen. The influence of the pollen’s aging and dehydration process on the pollenkitt are clearly visible. In (**e**), a pollen glued to an atomic force microscopy (AFM) cantilever is presented, showing that the pollen grain surface is left pristine by the gluing process.

**Figure 2 insects-13-00811-f002:**
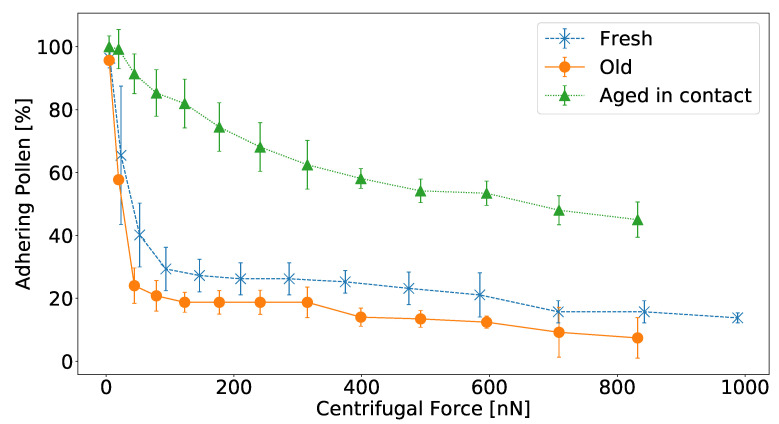
Quantification of *Hypochaeris radicata* pollen adhesion to glass using centrifugation. The amount of adhering pollen grains—relative to the number of pollen prior to centrifugation—is computed versus the centrifugal force. The adhesive properties were analyzed for pollen grains that were (1) fresh (blue crosses, n = 115); (2) old (orange circles, n = 165) when being deposited on the glass slide; and (3) were deposited freshly, but aged in contact with the glass prior to centrifugation (green triangles, n = 122). The mean values of four individual experiments are presented for each group of pollen and errorbars correspond to the standard deviation. Pollen that aged in contact with the glass adhered strongest to it. Fresh pollen grains had stronger adhesion than old ones.

**Figure 3 insects-13-00811-f003:**
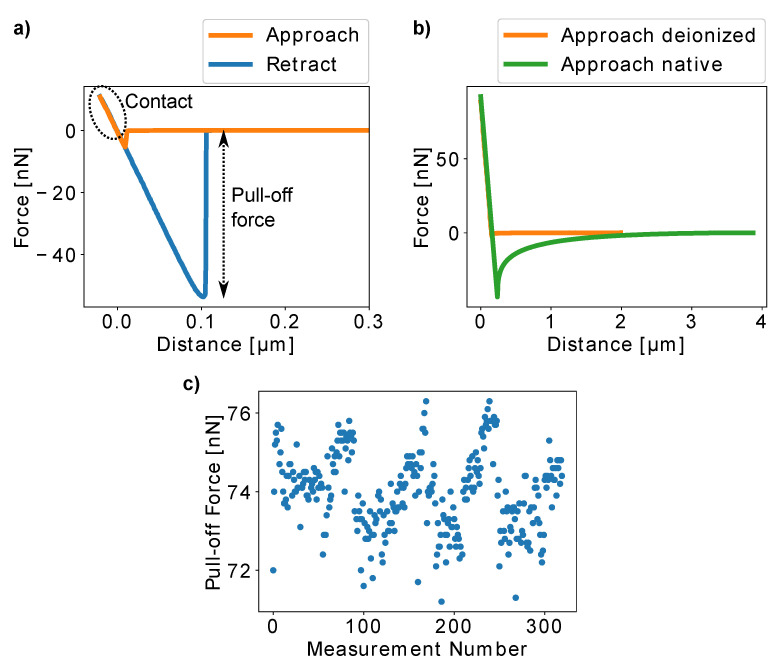
(**a**) An exemplary force–distance curve recorded with *Hypochaeris radicata* pollen on a glass slide. The curve’s linear behavior during the contact between pollen and glass indicates that the pollen was not deformed during the measurement, which ensures the repeatability of the experiment. The minimum of the retract part of the curve serves as a quantification of the adhesive forces between the pollen and the glass substrate (“pull-off force”). (**b**) The approach segments of force–distance curves before and after deionizing pollen and glass show that electrostatic interactions, which are sometimes present, can be removed from the setup using deionization. (**c**) 320 curves were recorded with one *H. radicata* pollen. The pull-off forces did not change significantly, showing that our experimental procedure is stable and repeatable.

**Figure 4 insects-13-00811-f004:**
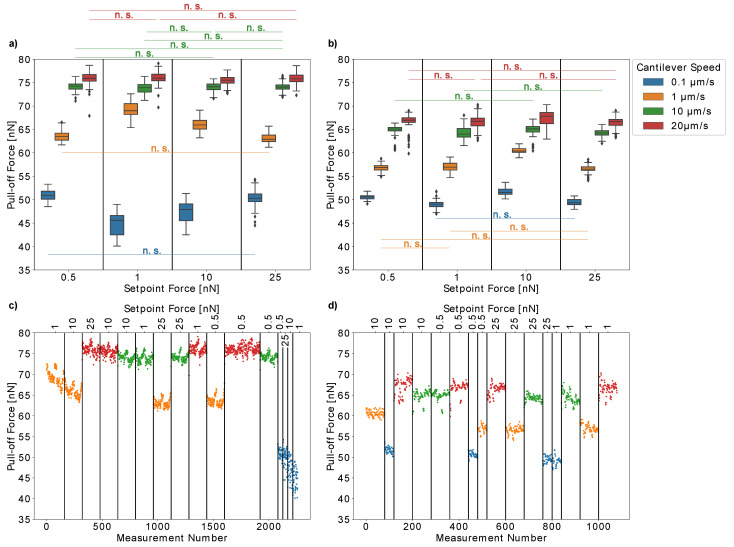
The influence of the setpoint force (contact force) and cantilever speed on the measured pull-off forces were analyzed for two fresh *Hypochaeris radicata* pollen. (**a**,**b**) The results presented as box plots. (**c**,**d**) The pull-off forces versus measurement number as an experimental control for the repeatability of measurements. For clarity, we only marked the boxes that are not significantly different from each other. Both individual pollen grains show an increase in pull-off forces with an increasing cantilever speed. The dependency on the setpoint force is not very pronounced. (**c**) The pull-off forces of the pollen presented in (**a**) decrease with a measurement number for low cantilever speeds, which could indicate that pollen aging influences our results. (**d**) The results obtained from the pollen presented in (**b**) were repeatable throughout the entire experiment. n.s.: no significance.

**Figure 5 insects-13-00811-f005:**
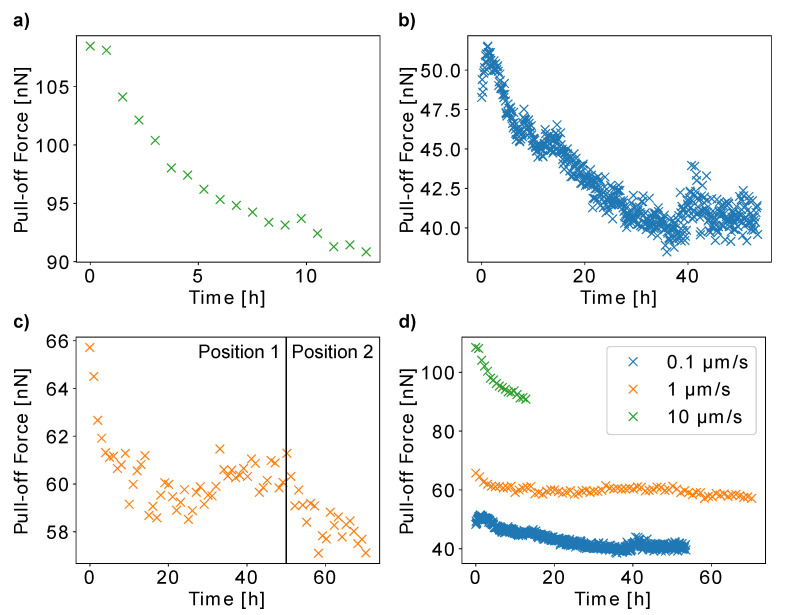
The pull-off forces of three single *Hypochaeris radicata* pollen grains from glass substrates measured repeatedly for 12 h (**a**) or more than 50 h (**b**,**c**) at different pull-off rates. (**d**) The pull-off forces plotted over time for all three pollen. The adhesiveness is decreasing over time for each pollen grain. The pollen measured with a cantilever speed of 10 µm/s has the highest pull-off forces, and the quickest decrease in adhesiveness.

**Figure 6 insects-13-00811-f006:**
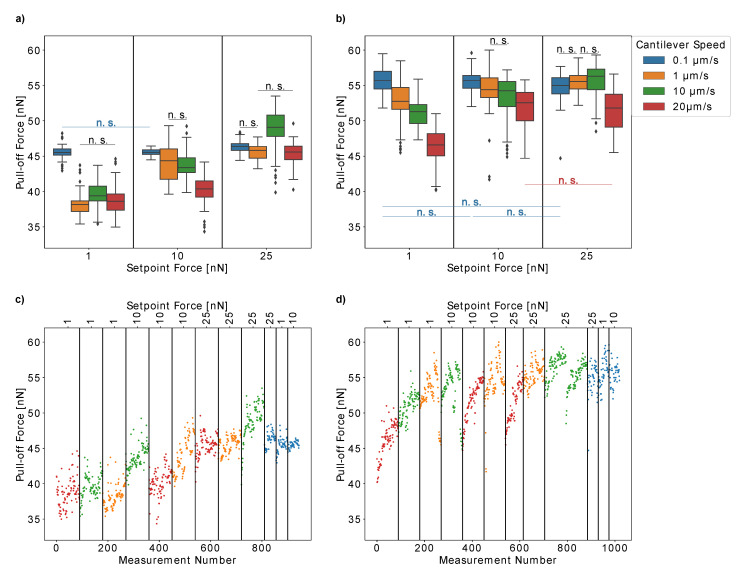
The influence of setpoint force and cantilever speed on pull-off forces for two individual pollen grains of *Hypochaeris radicata* that were older than seven days. (**a**,**b**) Box plots of the pull-off forces. (**c**,**d**) Pull-off forces versus measurement number. We marked only boxes that were not significantly different. In contrast to fresh pollen, the old pollen show a decrease in pull-off forces with increasing cantilever speed. For speeds above 1 µm/s, the pull-off forces increased with an increasing setpoint force. The pull-off forces for old pollen have a tendency to increase with the measurement number (**c**,**d**), which was not the case for fresh pollen. n.s.: no significance.

## Data Availability

The data presented in this study are openly available in FigShare at https://doi.org/10.6084/m9.figshare.20238342.v1.

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
