# Peer review of "Quantifying the Influence of Pollen Aging on the Adhesive Properties of Hypochaeris radicata Pollen"

_insects, 2022, doi:10.3390/insects13090811_

Round 1
Reviewer 1 Report
Title: Quantifying the Influence of Pollen Ageing on the Adhesive Properties of Hypochaeris Radicata Pollen
Authors: Steven Huth, Lisa-Maricia Schwarz and Stanislav N. Gorb
Overview: Authors Huth, Schwarz, and Gorb quantified the impact of aging on the adhesive properties of pollen (Hypochaeris radicata). The authors show that the adhesive properties of the pollenkitt, the most outer layer of pollen grains, changes over days—likely due to evaporation. The authors show the change in morphology of H. radicata pollen grains due to aging and the impact this has on both their adhesive properties as well as their ability to form new adhesions to hydrophilic surfaces. The authors show that pollen that ages while in contact with the artificial substrate, a coverslip, adhered more strongly to the glass relative to fresh or old pollen grains.
Problem: The adhesion properties of pollens as well as the influence of pollen aging and dehydration have not yet been quantified experimentally.
Working hypothesis: The adhesion forces of pollenkitt, quantified as the forces necessary to detach pollen from glass surfaces using centrifugation and a detachment process, will change over time.
Why should we care? A deeper understanding of the pollination process will provide insights into bioinspired passive microparticle manipulation through altering of surface properties, which is relevant for artificial pollination, drug delivery, respiratory medicine, and engineering of biomimetic microgripping systems.
General comments: Although I am not a botanist, I was sufficiently intrigued by the objective of this study to want to review the manuscript. I was not disappointed. The manuscript is beautifully written with clear, descriptive sentences. Methods were sufficiently detailed that replication by a different group of scientists could be conducted. Experimental trials were replicated throughout. The authors accounted for a number of confounding variables such as electrostatic interactions, location of pollen on the glass slide, and seasonal differences in the collection of pollen grains. Results were analyzed (almost to a fault with the force curve figures). The cryo-SEM images were spectacular and enhanced the idea that pollenkitt evaporates over time rather than shrinking by wear and tear from environmental particles. Conclusions were conservative. Future studies on the adhesion of pollen to more biologically relevant surfaces including the anther, stigma, and insects was discussed.
Specific comments: Line 340 has an extra word. Line 85, change “amount” to “number of” or “volume”.
Author Response
We would like to thank the reviewer for taking the time to read our manuscript and preparing a revision. We are glad that our work was perceived as intriguing and appreciate the positive feedback.

Reviewer 2 Report
The authors present novel and important results on time-dependend characteristics and the general attachment of pollen on glass substrates. Apart from my few concerns listed below, the study is clearly presented and highly interesting and, therefore, generally worth becoming published.
1) In the Materials section the authors write: „Flowering stems of Hypochaeris radicata (Asteraceae) were collected from open spaces around Kiel University (Kiel, Germany) between July and November“. I wonder if this rather broad season for pollen harvesting had an influence of pollen characteristics. Later on, the authors write: „Moreover, individual pollen grains might have already different ages when they are glued to the cantilever, because the time a blossom was already blooming before pollen collection varies as well“. Again, there is quite some uncertainty regarding the exact time-dependent characteristics of the tested pollen. I feel that the authors should discuss this uncertainty a little deeper as in the present form.
2) How long were pollen grains centrifuged? Did the experiments stop immediately when the target speed was reached? Pollen grains subject to higher target speeds were accordingly also subject to a longer mechanical load until the target speed was reached. How does this affect your results?
3) How was pollen deposited onto the glass slides? Was it poured out over the slides, or was it deposited with a tool? Do these different methods (probably) affect attachment?
4) Figure 2 legend states: „More than 110 individual pollen grains have been analyzed for each group.“ Would it not be beneficial to indicate the exact n for each trial?
5) I do not understand Figure 3c, which shows a continuous graph from „force over measurement number“ datapoints. How is that possible?
6) Generally, I admire the author’s methods and understand that especially the AFM measurements are time-consuming and difficult. However, since the AFM measurements are characterized by a (very) low n, I recommend to the authors to at least mention and discuss this a little deeper.
Author Response
We would like to thank the reviewer for the positive feedback and for taking the time to thoroughly read through our manuscript. We feel that the remarks and criticism helped a lot to significantly improve the quality of our manuscript.
